# A Novel Full-Digital Protocol (SCAN-PLAN-MAKE-DONE^®^) for the Design and Fabrication of Implant-Supported Monolithic Translucent Zirconia Crowns Cemented on Customized Hybrid Abutments: A Retrospective Clinical Study on 25 Patients

**DOI:** 10.3390/ijerph16030317

**Published:** 2019-01-24

**Authors:** Francesco Mangano, Bidzina Margiani, Oleg Admakin

**Affiliations:** Department of Prevention and Communal Dentistry, Sechenov First Moscow State Medical University, 119992 Moscow, Russia; margiani.b@gmail.com (B.M.); admakin1966@mail.ru (O.A.)

**Keywords:** intraoral scanners, individual hybrid abutments, monolithic translucent zirconia crowns, marginal adaptation, survival, success

## Abstract

Purpose: To present a novel, full-digital protocol for the design and fabrication of implant-supported monolithic translucent zirconia crowns cemented on customized hybrid abutments. Methods: The present retrospective clinical study was based on data from patients who had been treated with single Morse-taper connection implants (Exacone^®^, Leone Implants, Florence, Italy) and were prosthetically restored with monolithic translucent zirconia crowns, cemented on customized hybrid abutments. The full-digital protocol (SCAN-PLAN-MAKE-DONE^®^) consisted of 8 phases: (1) intraoral scan of the implant position with scanbody; (2) computer-assisted design (CAD) of the individual abutment (saved as “supplementary abutment design” in external folder) and temporary crown; (3) milling of the individual zirconia abutment and of the temporary polymethyl-methacrylate (PMMA) crown, with extraoral cementation of the zirconia abutment on the relative titanium bonding base, to generate an individual hybrid abutment; (4) clinical application of the individual hybrid abutment and cementation of the temporary PMMA crown; (5) two months later, intraoral scan of the individual hybrid abutment in position; (6) CAD of the final crown with margin line design on the previously saved “supplementary abutment design”, superimposed on the second scan of the abutment in position; (7) milling of the final crown in monolithic translucent zirconia, sintering, and characterization; and (8) clinical application of the final crown. All patients were followed for a period of 1 year. The primary outcomes of this study were the marginal adaptation of the final crown (checked clinically and radiographically), the quality of occlusal and interproximal contact points at delivery, and the aesthetic integration; the secondary outcomes were the 1-year survival and success of the implant-supported restoration. An implant-supported restoration was considered successful in the absence of any biological or prosthetic complication, during the entire follow-up period. Results: In total, 25 patients (12 males, 13 females; 26–74 years of age; mean age 51.1 ± 13.3 years) who had been restored with 40 implant-supported monolithic translucent zirconia crowns were included in this study. At delivery, the marginal adaptation was perfect for all crowns. However, there were occlusal issues (2/40 crowns: 5%), interproximal issues (1/40 crowns: 2.5%), and aesthetic issues (1/40 crowns: 2.5%). The overall incidence of issues at delivery was therefore 10% (4/40 crowns). At 1 year, one implant failed; thus the survival of the restorations was 97.5% (39/40 crowns in function). Among the surviving implant-supported restorations, three experienced complications (one loss of connection between the hybrid abutment and the implant, one decementation of the zirconia abutment, and one decementation of the zirconia crown). The success of restorations amounted to 92.4%. Conclusions: The restoration of single Morse-taper connection implants with monolithic translucent zirconia crowns cemented on customized hybrid abutments via the novel SCAN-PLAN-MAKE-DONE^®^ full-digital protocol seems to represent a reliable treatment option. However, further studies on a larger number of patients and dealing with different prosthetic restorations (such as implant-supported fixed partial prostheses) are needed to confirm the validity of this protocol.

## 1. Introduction

The intraoral scanner (IOS) represents an important innovation in the world of dentistry, able to transform conventional prosthetic [1,2], surgical [3], and orthodontic [4] workflows.

In fact, IOS allows one to capture impressions of the dental arches of patients without having to use conventional impression materials (polyether, polyvinylsiloxane, alginate) with relative trays, but using only a light source (a structured light grid or, more rarely, a laser) that is projected onto the teeth to be scanned [1,2,5]. The deformation that this light grid undergoes at impact with the teeth is captured by powerful cameras and reworked by the scanning software, which generates a cloud of points [2,5]. The point cloud is then triangulated to give the mesh, reconstruction of the surface of the scanned object, or the virtual model of the patient’s dental arches [2,5].

The advantages of using IOS are numerous: the superior comfort of the patient (patients have never loved the classic impressions with materials and trays, considered extremely annoying) [2,6,7,8,9]; the ease of use for the clinician even in complex cases [9,10]; the possibility of immediately obtaining digital models of the arches, available for evaluation, without having to go through the casting of a plaster model [11]. The plaster models are eliminated with the possibility of saving space in the dental office because the acquired files are saved in the computer and can be sent to the dental laboratory by e-mail, without additional costs [12].

To date, as reported in the literature, IOS can be successfully used in the prosthetic field for the capture of accurate impressions for the modeling and manufacturing of single crowns [13,14,15] or fixed partial prostheses (bridges up to 4–5 elements) [16,17], both on natural teeth and on implants, and for prostheses like Toronto bridges on 4 fixtures [18]; on the other hand, the use of IOS for the capture of impressions for the production of fixed full-arch prostheses on 6 and 8 implants is not yet supported by the literature [2,19,20,21].

Theoretically, in implant prosthodontics, IOS should find its ideal application. In fact, with dental implants there is no need to capture subgingival margins with light: the impression must exclusively detect the spatial position of the fixtures, through the acquisition of the scanbodies (devices for transferring the position of the implants, i.e., the modern version of impression transfers) [22]. In fact, within the chosen computer assisted design (CAD) software, the corresponding implant library can be recalled, on which the dental technician proceeds with modeling [22].

Thanks to the optical impression, it is possible to model and then mill the components necessary for prosthetic restoration on implants, whether screwed or cemented [2,23]. In particular, within the category of cemented restorations, there is the possibility for dentists to obtain from their dental technician individualized hybrid abutments (in titanium and zirconia), on which temporary restorations modeled and milled in polymethyl-methacrylate (PMMA) can be cemented; these temporaries are then replaced by definitive monolithic zirconia restorations [2,14,23]. Within this workflow, at least in theory, the optical impression with scanbodies in position could be used also for the design of the final prosthetic restorations, without having to capture a second impression. In the digital workflow, in fact, the implant position is imported into a CAD software, where the meshes of the implant scanbodies are replaced by corresponding library files, coupled to all the components of the implant producer (portion of the implant and above all, different titanium links/bases). Within the CAD software, the dental technician can therefore choose the link/base best suited to the context, and draw above it the upper portion of an individualized abutment; this upper portion is milled in zirconia, and cemented extraorally on the chosen titanium base [2,14,23]. In this way, the dental technician can prepare for the dentist an individualized hybrid abutment, made of titanium (lower portion) and zirconia (upper portion). The lower portion corresponds in fact to the chosen bonding base, a preformed component supplied by the implant producer, engaged into the implant; the upper part corresponds to the individual portion, modeled and then milled in zirconia by the dental technician, on which the temporary restoration in PMMA is cemented. Subsequently, the same CAD scene should be used by the dental technician for the design of the final restoration, to be milled in monolithic zirconia [2,14,23].

This workflow, dedicated exclusively to cemented prosthesis, is particularly elegant, and allows managing and individualizing different clinical situations, maximizing aesthetics through the customization of the abutment [14,23,24,25]. In fact, from the biological point of view, soft tissues interact optimally with zirconia abutments: after months adhesion and creeping on the zirconia is observed, with gingival regrowth, able to restore optimal aesthetics [26,27,28]. Similar results have not been reported with integral titanium abutments [28]. Furthermore, the color of zirconia (white) is more natural, and a possible transparency of the abutment does not have the negative effects that can occur with titanium (gray) [29].

Despite these undoubted advantages, within this workflow there are two critical moments that can cause errors, and therefore clinical problems. The first moment is that of the replacement of the mesh of the scanbody with the corresponding library file. If, in fact, the coupling managed through the CAD software is not perfect, and presents for example a small (horizontal) rotation, a position error can arise, so there will not be an exact correspondence between the position of the hybrid abutment in the virtual CAD scene and the actual position of the assembly in the mouth [14]. A second critical moment is then represented by the cementation of the customized portion of the abutment (in zirconia) on the preformed base (in titanium); this takes place extraorally and, if not performed correctly (for example, if the zirconia portion is cemented onto the base with a slight horizontal rotation with respect to the CAD design, due to the tolerances between the components), an error may arise, which will once again determine an inaccurate correspondence between the position of the hybrid abutment in the virtual CAD scene and the actual position of the assembly in the mouth [14].

Up to now, these two problems have represented for many colleagues a limit to the use of hybrid customized abutments in the cemented digital prosthesis. In fact, a second scan of the actual position of the zirconia abutments in the mouth, at the end of the provisional period, is not viable; the margins of the preparation of the abutments are not visible, since they are subgingival after tissue maturation. As a consequence, it can be impossible for the technician to precisely draw the margin line of the final restoration [14,22].

The most experienced dental technicians tried to solve this problem by scanning the physical piece/assembly of the individual hybrid abutment outside the mouth, with a desktop scanner, before sending it to the dentist for application. The .STL file of that scan can be saved, and used at the time of transition from a temporary to the final restoration, after receiving a second scan from the dentist (that of the zirconia abutments in the patient’s mouth). In fact, the dental technician can import the .STL file of the individualized hybrid abutment (with visible margins) in the final CAD scene derived from the second patient’s scan (where margins are not visible), replace it and then design the final restoration.

Although this approach can solve problems; however, it forces the technician to model on a mesh, derived by an extraoral scan (and not on an original CAD drawing); this can lead to errors in the identification of the margins of the restoration and to potential misfits between the final prosthesis and the customized hybrid abutment. In addition, it requires an additional step, i.e., scanning the hybrid abutment outside the mouth with a desktop scanner.

The aim of this article is to present a novel, patented, full-digital protocol (SCAN-PLAN-MAKE-DONE^®^) for the design and fabrication of fixed implant-supported monolithic translucent zirconia crowns cemented on customized hybrid abutments, without any additional desktop scan, but only with a second intraoral scan of the actual position of the abutments in the mouth.

## 2. Methods

### 2.1. Study Design

The innovative full-digital protocol (SCAN-PLAN-MAKE-DONE^®^) devised and registered by our research group, for the prosthetic rehabilitation of patients enrolled in this study, consisted of the following clinical and laboratory phases, as summarized in Figure 1:
SCAN1: intraoral scan of the implant position with scanbody;PLAN 1: design of the individual abutment from library files (bonding bases) and design of the temporary PMMA restoration; the files of the individual abutment (original CAD drawing) thus modeled are saved (as .STL files) in a specific folder, labeled as “supplementary abutment design”, ready to be recalled in the following phases:MAKE1: milling of the individual zirconia abutment and of the temporary PMMA crown, and subsequent extraoral cementation of the individual zirconia abutment on the relative titanium bonding base, to generate an individual hybrid abutment;DONE1: clinical application of the individual hybrid abutment and cementation of the PMMA temporary crown above it;SCAN2: intraoral scan of the individual hybrid abutment in the mouth, in position, after removing the temporary crown;PLAN2: in the definitive CAD scene, the mesh of the abutment in the correct position in the mouth is replaced by the .STL file of the individual abutment (“supplementary abutment design”, original CAD drawing) that was previously stored in a specific folder; the dental technician proceeds to model the final crown;MAKE2: milling of the final crown in monolithic translucent zirconia, sintering and characterization;DONE2: clinical application of the final crown in monolithic translucent zirconia.

### 2.2. Inclusion and Exclusion Criteria

The present retrospective clinical study was based solely on data from patients who had previously undergone implant therapy with the insertion of one Morse taper connection implant (Exacone^®^, Leone Implants, Florence, Italy) in the posterior areas (premolars and molars) of both arches (maxilla and/or mandible) in the period between June 2016 and June 2017, and which had been prosthetically rehabilitated in the same dental center through a full-digital protocol, without the use of physical models. This full-digital protocol consisted of the design and manufacture of monolithic translucent zirconia crowns, cemented on customized hybrid abutments. To be included in this retrospective study, the digital patient records/folders had to present all the clinical and radiological data necessary for enrollment: the intraoral scans captured for the design and manufacture of the individual hybrid abutment and for temporary and final restorations; the temporary and definitive CAD scenes; and the documentation of all the steps of prosthetic rehabilitation. Finally, to be enrolled, patients had to read and sign a document of adhesion to this retrospective study, on the nature of which they were informed in detail. On the contrary, all the patients who were rehabilitated through analogical prosthetic procedures (that is, with the capture of conventional impressions via materials and impression trays) were excluded from the present study, thus envisaging the development of plaster models, as well as all patients rehabilitated through mixed analog-digital protocols, and all the patients in which 3D-printed models were used. Finally, all patients restored through a full-digital protocol who nonetheless did not give consent for the inclusion in the present study were excluded. The present study was carried out in full compliance with the principles set out in the Helsinki Declaration on human experimentation of 1975 (and revision of 2008) and obtained the approval of the Local Ethics Committee at Sechenov University, Moscow, Russia.

### 2.3. Detail of Prosthetic Procedures

The prosthetic treatment of the patients proceeded according to the 8 different phases previously described, through the fully digital patented SCAN-PLAN-MAKE-DONE^®^ protocol. In particular, after a healing period following an implant placement of 2 to 3 months, the patient underwent a first optical impression with a powerful intraoral scanner (CS 3600^®^, Carestream Dental, Atlanta, GA, USA). The impression was captured by selecting, in the acquisition software, the module for capturing dental implants. The impression consisted of the capture of the antagonist arch, of the master model just after the healing abutment removal, and of the occlusal bite/registration; then continued with the placement of the scanbody, the capture of it, the check of the quality of the file, and the subsequent saving (Figure 2). Particular attention was paid to the capture of contact points with adjacent teeth (where present), bite, and the whole scanbody. The acquisition software allowed one to generate accurate and light meshes, to be saved in .STL format (or, with the color information embedded, PLY), ready to be imported into the selected CAD. Once the scans were imported into the CAD software (Exocad DentalCAD^®^, Exocad, Darmstadt, Germany), the mesh of the implant scanbody was replaced with the corresponding library file, with the selection of the different bonding bases (bases in titanium) for modeling. In particular, the library of implants used in this study (Exacone^®^, Leone Implants, Florence, Italy) envisaged the choice between a straight titanium base 4 mm in height (Tibase^®^, Leone Implants, Florence, Italy) or, alternatively, a 6-mm-high titanium base (Multitech^®^, Leone Implants, Florence, Italy) available in either a straight or an angled version (with angle of 15°). For maximum prosthetic versatility, in the case of the need to correct implant angles, the angled abutments library provided the possibility of rotating horizontally the titanium bases 30° in 30° (12 positions instead of 6). Having chosen the bonding bases on which to design, the individual zirconia abutment was modeled, with the prosthetic superstructure of the PMMA provisional crown (Figure 3 and Figure 4). The file of the individual hybrid abutment (original CAD drawing) was therefore saved separately in a dedicated folder as a .STL file, labeled as “supplementary abutment design” (to be recovered in the subsequent modeling phase) (Figure 5). The cement space was set at 0.1 mm. Then the computer-assisted-manufacturing (CAM) phase started and the zirconia individual abutment was milled with a powerful 5-axis milling machine (Roland DWX-50^®^, Roland Easy Shape, Ascoli Piceno, Italy), sintered (Tabeo^®^, Mihm-Vogt, Stutensee, Germany), and cemented extraorally on the selected titanium base, according to the CAD project. The provisional crown was instead milled in PMMA using a 4-axis milling machine (Roland DWX-4^®^, Roland Easy Shape, Ascoli Piceno, Italy). Upon delivery of the provisional restoration, the individual hybrid abutment was activated with a percussion hammer and the PMMA crown was cemented using a zinc-oxide eugenol cement (TempBond^®^, Kerr, Orange, CA, USA). After a careful check of the perfect occlusal adaptation and congruency of the contact points, polishing and characterization were done. Articulating papers (Bausch Articulating Paper^®^, Bausch Inc., Nashua, NH, USA) were used to check the occlusion, and gingival floss was used to verify the quality of contact points. The provisional restoration remained in situ for a period of 2 months (Figure 6), after which they were replaced by the definitive monolithic zirconia restoration. In fact, after completing the provisional period, useful to verify the adaptation of the implant under prosthetic load, the patient was recalled for a further optical impression of the dental arches that was performed with the same aforementioned intraoral scanner (CS 3600^®^, Carestream Dental, Atlanta, GA, USA). This time, however, the impression was made after removing the temporary restoration, in order to capture the actual position of the individual zirconia abutment in the mouth (Figure 7). The acquisition method of natural teeth was used. The scan of the antagonist, the master model, and the bite was performed quickly, and it was imported into the CAD software (Exocad DentalCAD^®^, Exocad, Darmstadt, Germany). In the CAD software, the clinician replaced the mesh of the abutment with the previously saved .STL file of the individual hybrid abutment design (original CAD drawing, “supplementary abutment design”). This was done in order to allow a correct visualization of the margins of the abutment, otherwise not visible because subgingival, and to allow the operator to design the final restoration on an original CAD drawing (with perfect margins) instead of on a mesh derived from an intraoral scan. In short, the “supplementary abutment design” .STL file saved from the previous CAD scene was imported into the new CAD scene as an “additional scan.” This mode allowed us to replace the scan mesh with the original CAD drawing, with perfectly designed and visible margins. In other words, the original CAD drawing of the individual hybrid abutment was superimposed on the mesh of the abutment itself, and replaced it (Figure 8 and Figure 9). The technician could therefore proceed to draw the margin line without difficulty, on a perfect CAD drawing (and not on an imperfect mesh), positioned correctly (in the same actual position as the abutment in the mouth). The software was able to automatically detect the margin lines (Figure 10). The quality of the superimposition was verified mathematically with a powerful reverse-engineering software (Studio2012^®^, Geomagics, Morrisville, NC, USA) that revealed little deviations between the two surfaces (Figure 11), confirming the quality of the overlapping through the algorithms of Exocad^®^. In a few minutes it was possible to draw the final restoration (Figure 12). The cement space in this case was between 0.07 and 0.08 mm. The definitive monolithic crown in translucent zirconia was produced by milling with a powerful 5-axis milling machine (Roland DWX-50^®^, Roland Easy Shape, Ascoli Piceno, Italy), subsequently sintered in an oven (Tabeo^®^, Mihm-Vogt, Stutensee, Germany), characterized, and ready for cementation. On delivery, the clinician carefully checked the marginal adaptation with magnifying glasses (Zeiss 4.5x^®^, Zeiss, Oberkochen, Germany) and a periodontal probe. The goodness of the contact points with adjacent teeth (if present) was carefully checked with dental flosses and the absence of occlusal precontacts was controlled with the aid of articulating papers (Bausch Articulating Paper^®^, Bausch Inc., Nashua, NH, USA). The clinician reported in the patients’ electronic records/folders all possible problems and issues encountered at the delivery of the final crowns. Finally, cementation was carried out with the same zinc-oxide eugenol cement (TempBond^®^, Kerr, Orange, CA, USA) used to cement the temporary restorations (Figure 13A). An endoral periapical radiograph was also taken (Figure 13B).

### 2.4. Outcome Variables

There were five outcome variables in the present study. The first three outcomes (primary outcomes) were evaluated at the delivery of the final restorations, whereas the last two (secondary outcomes) were evaluated 1 year after the delivery of the final crowns. Accordingly, the first three (primary) outcomes of the present study were: (1) the marginal adaptation of the final crown, which was checked clinically with the aid of magnifying glasses and radiographically with an endoral periapical radiograph; (2) the quality of occlusal and interproximal contact points; and (3) the aesthetic integration of the final monolithic translucent zirconia crown. Moreover, all patients were included in a recall control program that included at least two professional oral hygiene sessions per year. During these sessions, the clinician could check the status of the restoration and possibly note in the electronic medical record/folder the presence of any problem or complication occurring during the follow-up period. The clinician had therefore the possibility to evaluate the last two (secondary) outcomes of this study: (4) survival and (5) success of the implant-supported restoration.

#### 2.4.1. Marginal Adaptation of the Final Restoration

The marginal adaptation of the final restoration was carefully checked by the clinician at the delivery of the monolithic translucent zirconia crown. The marginal adaptation was checked clinically using magnifying glasses (Zeiss 4.5x^®^, Zeiss, Oberkochen, Germany) and phisically probing the margin area with a periodontal probe. This procedure was performed circumferentially all around the crown in order to intercept any possible misfit, gaps, or undercuts. Finally, a radiograph of the implant-supported restoration was taken to radiographically check the perfect seating and marginal adaptation of the final restoration. If the marginal adaptation was satisfactory, the clinician could proceed to cement the final crown; conversely, if the marginal adaptation was unsatisfactory, the crown was not cemented and the work was sent back to the dental technician for re-evaluation. In this case, anyway, the work had to be completely remade.

#### 2.4.2. Quality of Occlusal and Interproximal Contact Points

The occlusion check, using articulating papers (Bausch Articulating Paper^®^, Bausch Inc., Nashua, NH, USA), was particularly accurate. In the case of small boosters, the crown was polished directly in the dental clinic and so could be applied. In that case, however, the precontacts were excessive and the shape of the occlusal surface was not satisfactory, the crown was not cemented, and the work was sent back to the dental technician for re-evaluation (or remaking). Similar considerations were valid for the contact points with the adjacent teeth, which, if present, necessarily had to be satisfactory. The control of the contact points with the adjacent teeth was done using dental flosses. If the contact points were not satisfactory, in order to avoid stagnation of food and hygienic problems the crown was not cemented and was sent back to the laboratory for remaking.

#### 2.4.3. Aesthetic Integration of the Final Crown

The aesthetic integration of the restoration was evaluated at the time of delivery and was considered satisfactory if the final crown had a color similar to that of the adjacent teeth and therefore was able to mimic them. If the aesthetic integration was considered satisfactory, the clinician proceeded to cement the final restoration. In case it was not considered satisfactory, the crown was sent back to the technician for further characterization.

#### 2.4.4. Survival of the Implant-Supported Restoration

An implant-supported restoration was classified as “surviving” if it was still present in the mouth at the time of the final check-up visit, i.e., 1 year after the insertion of the final crown [30,31]. On the contrary, it was considered “failed” if it was necessary to make it up again. The causes of implant-supported restoration failure were: (1) loss of the implant; and (2) fracture of the crown such that it cannot allow anything other than its replacement [30].

#### 2.4.5. Success of the Implant-Supported Restoration

A prosthetic restoration was classified as “successful” if it did not present any complication during the entire observation period [31] from delivery to control at 1 year. Otherwise, if even one of the possible complications occurred, the restoration was classified as “unsuccessful.” Among the complications, biological and prosthetic complications were considered [32]. The biological complications were: (1) peri-implant mucositis with gingival swelling, discomfort, and/or bleeding [33]; and (2) peri-implantitis with pain, suppuration, bleeding, and/or marginal bone resorption [34]. The prosthetic complications were: (1) mechanical, i.e., affecting pre-formed components sold by the implant manufacturer (implant, bonding base) such as the loss of connection between the abutment and implant or the fracture of the fixture or the bonding base; and (2) prosthetic, i.e., affecting the components designed and manufactured by the dental technician (individual zirconia abutment, monolithic crown in translucent zirconia), such as decementation of the upper portion of the abutment, fracture of the upper portion of the abutment, and decementation or chipping of fracture of the monolithic translucent zirconia crown [32,35].

### 2.5. Statistical Evaluation

All relevant data was collected from the electronic medical records/folders of the patients by an independent experienced operator who was not directly involved in the placement of fixtures nor in the prosthetic restoration of the patients. Descriptive statistics were performed for the patients’ demographics (gender, age at start of the prosthetic treatment) and the location/position of the crowns. Absolute distributions were calculated for qualitative variables (marginal adaptation, quality of occlusal and interproximal contacts, aesthetic integration, survival, complications, success) while means, standard deviations, medians, and 95% confidence interval (CI) were estimated for quantitative variables (patient’s age at start of the prosthetic treatment). The survival and success of the zirconia crowns and the incidence of complications were calculated at the restoration level.

## 3. Results

In total, 25 patients (12 males, 13 females; age range 26–74 years of age, mean age 51.1 ± 13.3 years, median age 48 years; CI 95% 45.9–56.3 years) who had been restored with 40 implant-supported monolithic translucent zirconia crowns (25 maxilla, 15 mandible; 12 premolars, 28 molars) were included in this study. Among these patients, only 6 were smokers. The restorations present in this study were supported by zirconia abutments cemented on different titanium bases (8 Tibase^®^, 10 Multitech^®^ straight, and 12 Multitech^®^ angled 15°), which were retained through a Morse taper connection inside fixtures of different diameters (one 3.3-mm implant, 15 4.1-mm implants, 20 4.8-mm implants, and four 5.5-mm implants) and different lengths (four 6.5-mm implants, ten 8-mm implants, 14 10-mm implants, eight 12-mm implants, and four 14-mm implants).

In all cases (40/40 crowns, 100%) the marginal adaptation was optimal as clinically (inspection with magnifying glasses and probing) and radiographically (with periapical endoral radiograph) verified. Occlusal adaptation, on the other hand, presented some problems. In fact, in 5 cases (5/40, 12.5%) the occlusion of the definitive restorations was not perfect and had to be retouched by polishing the cusps before cementation; in two other cases (2/40: 5%) the crowns had to be sent back to the dental technician, remodeled, and milled again, as the occlusal precontacts were such that it was not possible to polish and apply them. Interproximal adaptation was instead optimal in all cases, except for a crown (1/40: 2.5%) which had rather weak and loose contact points and was therefore sent back to the dental technician for remodeling and milling. Finally, from an aesthetic point of view, the integration of monolithic translucent zirconia restorations was quite satisfactory, with only two cases of non-perfect chromatic integration; in one of these (1/40: 2.5%); however, the crown was sent back to the dental technician for further characterization. In the other case, in fact, the patient was still satisfied with the color and asked for the final restoration to be cemented without further chromatic changes, or delay. In summary, therefore, only four monolithic translucent zirconia crowns (4/40: 10%) were sent back to the dental technician (2 for insufficient occlusal adaptation, 1 for insufficient interproximal adaptation, and 1 for insufficient aesthetic integration); 3 of these had to be completely redone (therefore redesigned and milled) and 1 was differently characterized. In any case, the dental technician corrected the errors and the imperfections and the new restorations, reviewed and corrected, could be subsequently cemented. The problems encountered at the delivery of the final crowns are summarized in Table 1.

During the follow-up period there were no dropouts, because all patients regularly presented themselves at the two annual check-ups for the professional oral hygiene session. At the end of 1 year, only one crown (in the maxillary molar area) was lost (because the patient lost the implant). This implant failure, which occurred at 2 months from the positioning of the final crown, occurred in the absence of infection; the patient was a smoker and the implant supporting the restoration was extra-short (6.5 mm). No other failures occurred; hence, the survival of the restorations at 1 year was 97.5% (39/40 crowns in operation). Similarly, in the group of patients selected for this study, there were no biological complications (i.e., peri-implant mucositis and/or peri-implantitis). However, complications of a prosthetic nature were more frequent, affecting 3 crowns (3/39 functioning: 7.6%). In a single patient there was a loss of connection between the hybrid abutment and the Implant. This abutment was repositioned and reactivated through axial percussion and did not present any more problems at the end of the study. In another patient, the upper portion of the hybrid abutment (part in zirconia) decemented from the bonding base (Tibase^®^); this required a new extraoral cementation of the individual abutment on the titanium base. This procedure was performed after careful cleaning and sandblasting of the base (to increase the adhesion surface). Finally, one zirconia crown decemented from the hybrid abutment. In this case, recementing it sufficed. Globally, these three complications did not involve particularly long or difficult interventions by the clinician, and could be resolved quickly. In any case, the success rate of implant-supported restorations was 92.4% (36/39 restorations were in function and did not present complications throughout the study period). The failures and complications found during the follow-up period and at the end of the present study—subdivided according to patient characteristics, type of bonding base, and type of implant—are reported in Table 2.

## 4. Discussion

The restoration of implants through customized zirconia abutments is one of the highest, most elegant, and most modern expressions of digital prosthodontics [12,16,17,26,27,28]. In fact, it is now possible for the dental technician to model individual abutments of ideal shape to support prosthetic restorations; moreover, the aesthetics of these restorations is particularly high and the material used (zirconia) is particularly pleasing to the soft tissues [17,26,27,28].

However, despite the remarkable advances made by intraoral scanners in terms of trueness and precision (and therefore accuracy) [2,23,36], the full-digital workflow for implant restoration through monolithic translucent zirconia crowns supported by customized hybrid zirconia titanium abutments still has critical issues.

If, for instance, the transfer of the implant position from actual to virtual (i.e., the scanbody) is positioned correctly, the error that occurs during acquisition with intraoral scanner is rather limited in the case of single crowns, as shown in several scientific articles [2,23,36,37,38,39]. However, it is during the subsequent CAD and CAM phases that more serious errors can occur, compromising the final result. In the CAD phase, in particular, the most delicate moment is at the replacement of the scanbody mesh, captured with the intraoral scanner, with the original library files provided by the implant manufacturer. In fact, if the dental technician does not perform the superimposition carefully, errors and rotations can occur between the parts; inevitably, the technician will model his own customized abutment (and therefore his restoration) in an incorrect position with respect to the actual one—this without considering possible errors in the design of the implant libraries (not all implant companies today are digital-ready). Further errors can then occur during the CAM phase, especially if low-performance milling machines are used and the milling strategies are not dedicated to the implant system in use [40]; but the greatest risk is represented by the extraoral cementation of the upper portion of the abutment (in zirconia) on the selected bonding base (titanium). In fact, in order to cement the two parts together, it is necessary that there be a certain tolerance between them. A cementation not in axis and with the portions rotated among them, how ever slightly, can cause problems of position. The use of 3D-printed models could help, but unfortunately the need to use digital analogues, physical pieces to insert in a printed hollow model, can introduce other problems, as different printers (and different resins) have different dimensional tolerances and little is known about the effective accuracy of these models [41].

In summary, when working with individual hybrid abutments, the sum of all these errors can result in a discrepancy between the planned CAD position of the abutment and the actual position of the abutment in the mouth, with potential problems in the transition from the provisional to the final restoration. This may be true for any implant system.

A possible strategy to overcome these errors has been found by the most experienced dental technicians, who scan the hybrid abutment outside the mouth with a powerful desktop scanner, immediately after extraoral cementation and before sending it to the dentist for application; then they save this mesh in a specific folder and retrieve it when designing the final restoration. In fact, they ask the clinician to perform a second scan of the abutments in the mouth, after the provisional period has ended. Although in this second scan the prosthetic margins of the abutments are not visible because subgingival, advanced CAD tools allow the dental technicians to accurately replace the same abutments with the meshes previously captured via desktop scanner (in which the prosthetic margins are visible). At this point the dental technician can model, in a second CAD, the definitive restorations, starting from the correct position of the abutments and with clearly visible margins (though subgingival).

Although this technique is deserving, and allows one to solve the problems mentioned above, it itself has some critical issues: it forces the technician to perform additional desktop scans (one per each customized hybrid abutment) and, above all, forces the technician to model on a mesh and not on an implant library file (as it should be). This is certainly the main problem, since a mesh is always a geometric approximation of reality, with inherent geometric-dimensional limits [2]. The fact that the technician is therefore forced to model on a mesh, as if modeling on a natural tooth (and not on an implant library), is undoubtedly a limitation of this approach.

The best possible approach is certainly to model the final restoration on library files, in order to have a clear, easy-to-read margin line, to facilitate the dental technician in the CAD design, the machines that produce in the CAM phase, and ultimately to improve the marginal closure of the restorations on the abutments. This is why we have studied and present here a new full-digital protocol (SCAN-PLAN-MAKE-DONE^®^) for rehabilitation on implants with monolithic translucent zirconia crowns, cemented on customized hybrid abutments, which overcomes this problem and which moreover is perfectly suited to the implant system used here. The implant system used in this study (Exacone^®^, Leone Implants, Florence, Italy) is in fact characterized by a locking-taper screwless implant-abutment connection [30,32,42,43,44]. In this system, all the bonding bases (Tibase^®^ 4 mm, and Multitech^®^ 6 mm straight and 15* angled) have an apical index (an hexagon that is useful only for repositioning the abutment) [14]. However, in the case of angled Multitech^®^, this hexagon is removable, due to the fact that the implant library has 12 positions (and not 6). This gives excellent versatility in CAD, but it can lead to difficulties in positioning that can be associated with possible errors—in scanning, in overlap in CAD, and, not least, in precision—during extraoral cementation of zirconia over titanium. It should also be noted that, in Morse taper screwless implants, the abutment is usually not removable once placed in the mouth (according to the one abutment—one time concept) [45]. This is positive because it allows the soft tissues an optimal healing on zirconia, avoiding the disruptive phenomena at the mucosa interface related to the repeated removal of the abutment [46]; yet it can create problems in cases of positioning errors.

Our approach basically consists in saving as a .STL file the modeling of the hybrid abutment made in the first CAD, and in re-importing this file (a library file with the preparation margins of the abutment clear and visible) in the second CAD, derived from the second intraoral scan (the one with the abutments in the mouth and in the correct position, captured at the end of the provisional period), so as to replace a scanning mesh (but in the correct position) with a library file. This can be done simply through CAD operators such as the import of “additional scan” into Exocad^®^. At this point the library file, with clear and visible margins, is superimposed on the scanning mesh, and replaces it. A model is therefore obtained in which it is possible to clearly visualize subgingival margins, and the technician can easily model the final restoration on a library file. The whole procedure is based on the powerful superimposition algorithm of the Exocad^®^ software, mathematically verifiable with powerful, additional reverse engineering software.

The present retrospective study (which is, to the best knowledge of the authors, unique in kind) included 25 patients (12 males, 13 females; ranging between 26 and 74 years of age, mean age 51.1 ± 13.3 years, median 48 years; CI 95% 45.9–56.3 years) who had been restored with 40 implant-supported monolithic translucent zirconia crowns (25 maxilla, 15 mandible; 12 premolars, 28 molars).

At delivery, the marginal adaptation was perfect for all zirconia crowns (40/40: 100%). This seems to validate the present CAD superimposition protocol. However, there were occlusal issues (2/40 crowns: 5%), interproximal issues (1/40 crowns: 2.5%), and aesthetic issues (1/40 crowns: 2.5%). The occlusal and interproximal issues may be mainly related to mistakes during the CAD process of the final crowns, rather than being determined by dimensional variations of zirconia during sintering. Since these issues represent major problems—problems that force the clinician to send the restoration back to the dental technician for re-making, with (high) costs—appropriate solutions should be found and the possibility of testing the occlusal and interproximal adaptation of the definitive design via a try-in crown milled in low-cost material (polyurethane) should be considered. This polyurethane try-in is tested directly in the mouth and can also be useful for verifying marginal adaptation. Once the marginal adaptation, occlusion, and contact points have been validated, this polyurethane try-in crown is eliminated and the corresponding design is used to mill the final zirconia restoration. In this study, moreover, one monolithic zirconia crown failed to aesthetically integrate in the mouth of patient. With monolithic, non-stratified crown milled from zirconia discs and then painted/characterized manually, aesthetic integration is still a issue; it must be pointed out that translucent zirconia is aesthetically better than a non-translucent one; and in any case this issue will probably be solved with the advent of ceramic 3D-printing technology. Overall, therefore, the incidence of issues at delivery was 10% (4/40 crowns). This can be considered rather high, and efforts should be made to reduce it. The failures and complications encountered in the follow-up period were, however, rather low. At 1 year, one implant failed; therefore, the survival of the restorations was 97.5% (39/40 crowns in operation). Among the surviving implant-supported restorations, three experienced complications (one loss of connection between the hybrid abutment and the implant, one decementation of the zirconia abutment, one decementation of the zirconia crown). These complications were considered minor in nature because they did not require long interventions by the clinician. In accord with the definition of success established for this study (i.e., absence of complications during the 1-year follow-up period), the rate of complications encountered in this study was rather low (7.6%) and the success rate of restorations was 92.4%. These results are in accordance with the previous literature, which reports excellent survival and success rates for monolithic implant-supported zirconia crowns in the short term [47,48].

Naturally, the present study has its limits. First is the limited number of patients enrolled, besides the fact that they were restored with single crowns and not with more complex restorations like partial fixed prostheses. To be able to draw more definitive conclusions on the validity of this method, studies on a larger number of patients are necessary, and with short- and long-span fixed partial prostheses. For more complex applications, it might perhaps be appropriate for the abutments in the first CAD phase to be drawn with geometric shapes impressed, so as to facilitate superimposition during the second CAD phase. Finally, the present study is retrospective in nature and it is well known that only by prospective clinical trials and randomized clinical trials we can draw more definitive conclusions about the validity of new protocols.

## 5. Conclusions

The restoration of single Morse-taper connection implants with monolithic translucent zirconia crowns cemented on customized hybrid abutments via the novel SCAN-PLAN-MAKE-DONE^®^ full-digital protocol seems to represent a reliable treatment option, with an excellent marginal adaptation and a rather low incidence of failures (for an overall survival rate of 97.5%) and complications (for an overall success rate of 92.4%) one year after delivery. Still, some issues linger, regarding occlusal, interproximal, and aesthetic integration of the monolithic restorations at delivery. Further studies on a larger number of patients and dealing with different prosthetic restorations (such as implant-supported fixed partial prostheses) are needed to confirm the validity of this protocol.

## Figures and Tables

**Figure 1 ijerph-16-00317-f001:**
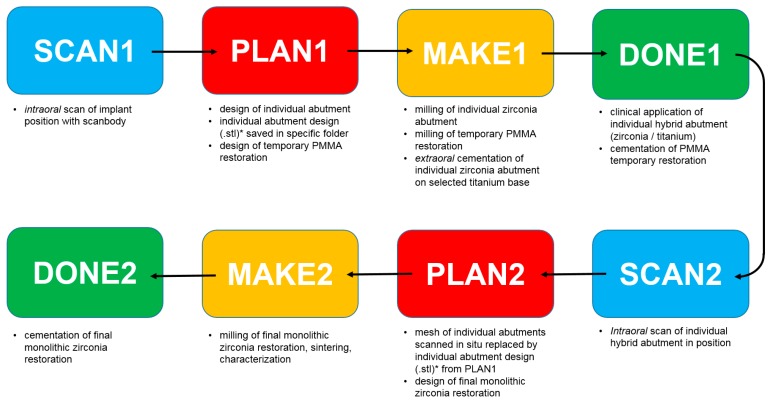
Schematic drawing of the patented full digital protocol (SCAN-PLAN-MAKE-DONE^®^) used in the present retrospective clinical study on 25 patients.

**Figure 2 ijerph-16-00317-f002:**
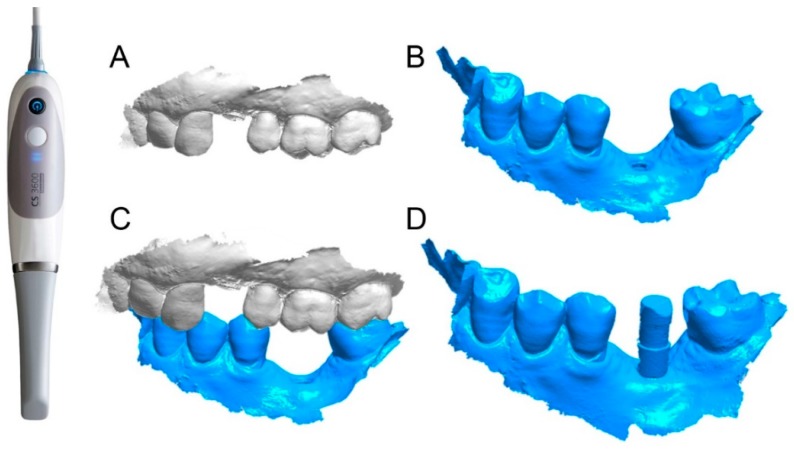
First intraoral scan (SCAN1) of the implant position (#36) with CS 3600^®^ (Carestream Dental, Atlanta, GA, USA). (**A**) antagonist arch; (**B**) master model; (**C**) occlusion; (**D**) scanbody in position.

**Figure 3 ijerph-16-00317-f003:**
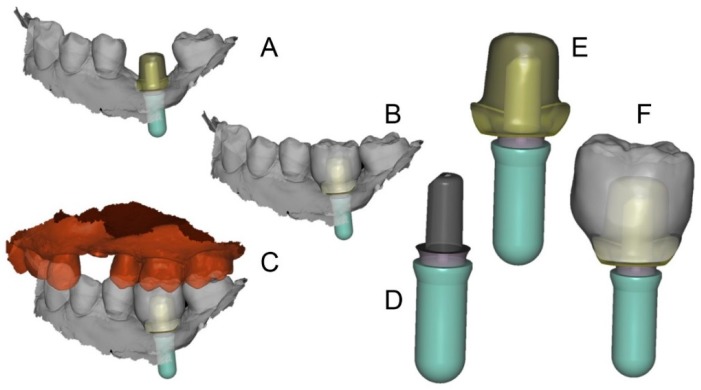
First CAD scene (PLAN1) with Exocad DentalCAD^®^ (Darmstadt, Germany). (**A**) the individual hybrid abutment in position; (**B**) the individual hybrid abutment and the provisional crown in position; (**C**) bite registration; (**D**) in this case, the selected titanium base was a Multitech^®^ straight (Leone implants, Florence, Italy); (**E**) the individual abutment has been modeled; (**F**) details of the assembly.

**Figure 4 ijerph-16-00317-f004:**
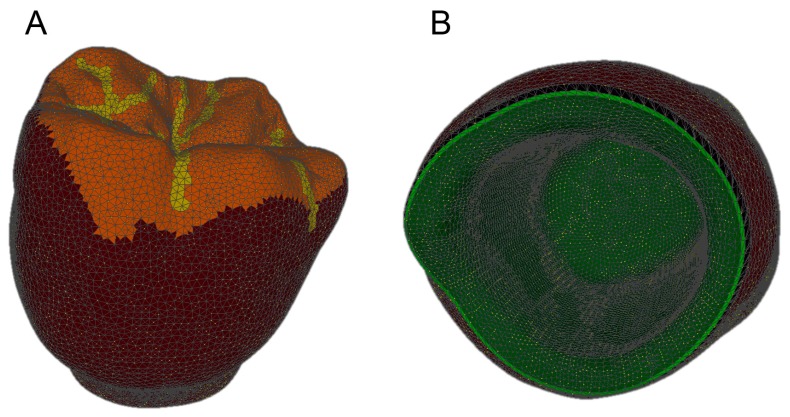
The provisional crown (PLAN1) as visualized in an open-access software (Meshlab^®^, Pisa, Italy). (**A**) external view of the mesh; (**B**) internal view of the mesh with the margin line clearly visible (in implantology, the margin line is designed on a library file, and not on a mesh, therefore it is perfect).

**Figure 5 ijerph-16-00317-f005:**
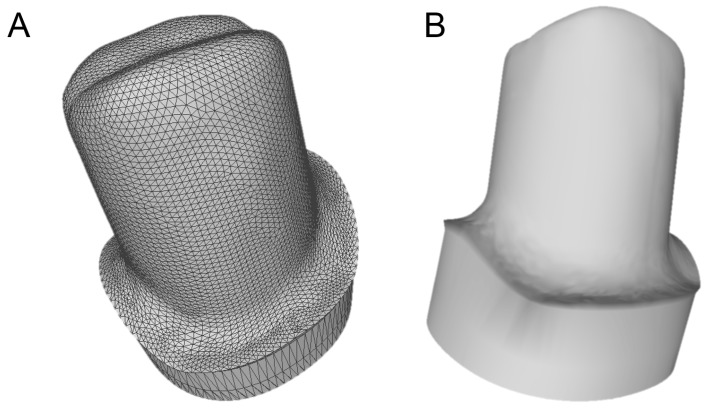
The file of the individual hybrid abutment designed during PLAN1 is saved separately in a dedicated folder as a .STL file (ready to be recovered in the subsequent modeling phase, or PLAN2). (**A**) the mesh with triangles in evidence; (**B**) the integral abutment design.

**Figure 6 ijerph-16-00317-f006:**
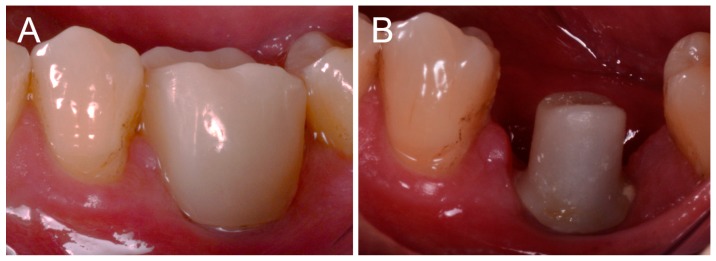
After 2 months of provisionalization, the patient is ready for the final scan. (**A**) the provisional PMMA crown in situ after a 2-month period; (**B**) The zirconia abutment immediately after the removal of the PMMA crown.

**Figure 7 ijerph-16-00317-f007:**
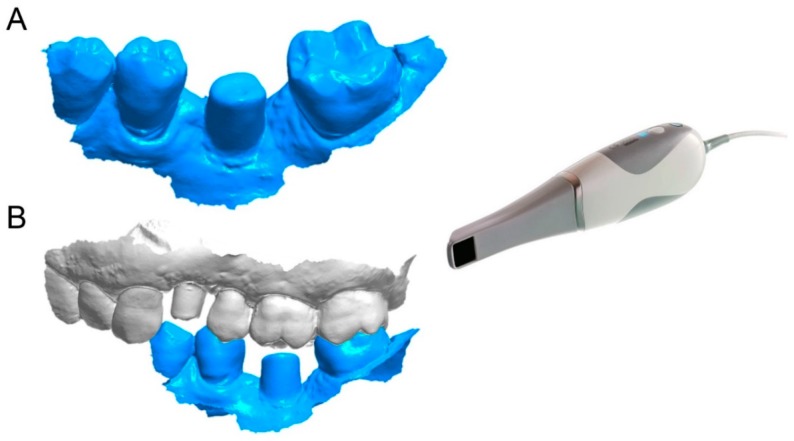
Second scan (SCAN2) with CS 3600^®^ (Carestream Dental, Atlanta, USA) at the end of the provisionalization. (**A**) the actual position of hybrid abutment position is captured intraorally, without taking care of the margin line that is clearly subgingival; (**B**) the relationship with the opposing arch.

**Figure 8 ijerph-16-00317-f008:**
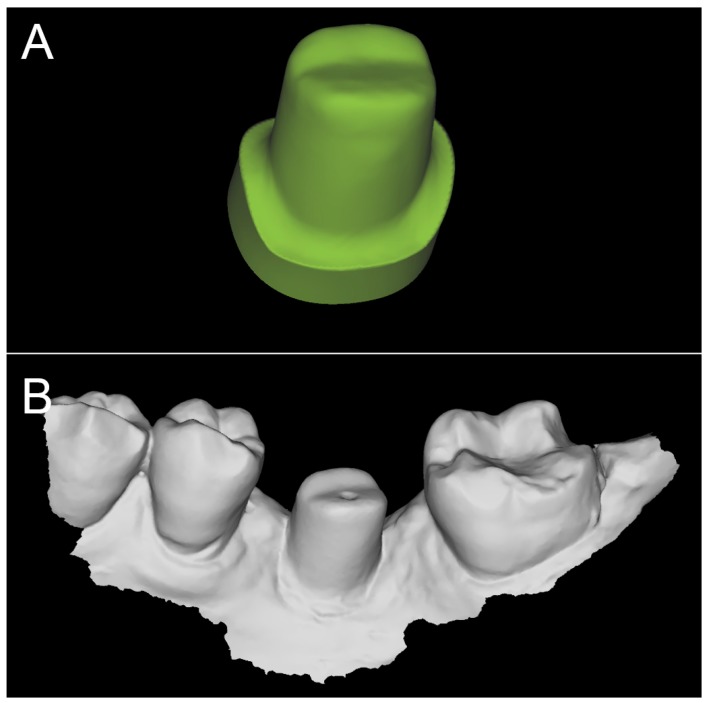
The original CAD drawing is imported in the CAD software for the superimposition. (**A**) the original CAD drawing of the individual abutment with clear and visible margin lines; (**B**) the corresponding mesh captured in the mouth, in which the margin line is not visible because subgingival.

**Figure 9 ijerph-16-00317-f009:**
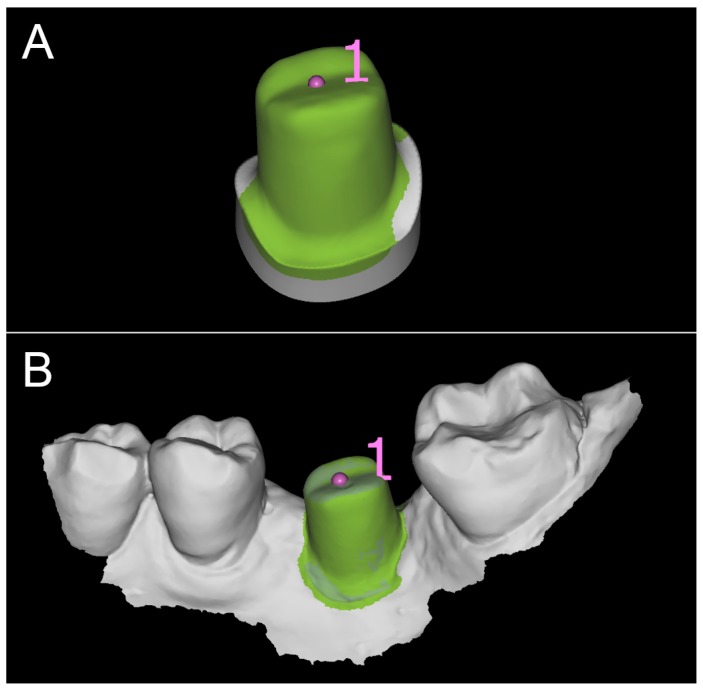
Superimposition. (**A**) the mesh captured in the mouth is replaced by the original CAD drawing. The superimposition proceeds first per points and then per surfaces; (**B**) The mesh (a geometric approximation of the object) is replaced by the original CAD drawing, so that the subgingival margins of the abutment become visible.

**Figure 10 ijerph-16-00317-f010:**
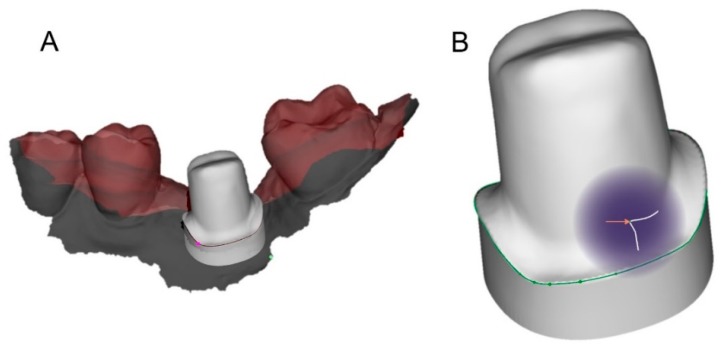
Second CAD scene (PLAN2) with the design of the final restoration. (**A**) the margine line of the individual hybrid abutment is clearly visible, even if subgingival, and therefore automatically detected by the software; (**B**) the dental technician can always control the margin line and modify it.

**Figure 11 ijerph-16-00317-f011:**
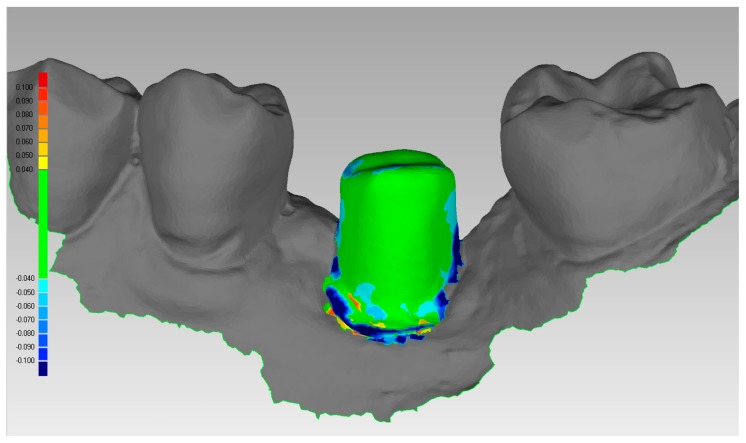
Mathematical control of the quality of the superimposition obtained with Exocad^®^. The files of the drawing of the individual hybrid abutment, from PLAN1, and the second scan (SCAN2), superimposed with the powerful algorithms of Exocad, are saved in that position and imported in a powerful reverse-engineering (Studio2012^®^, Geomagics, Morrisville, NC, USA). This software is used to calculate the distance/difference between the two files. A color-coded scale is generated, that revels little distances/deviations between the two surfaces.

**Figure 12 ijerph-16-00317-f012:**
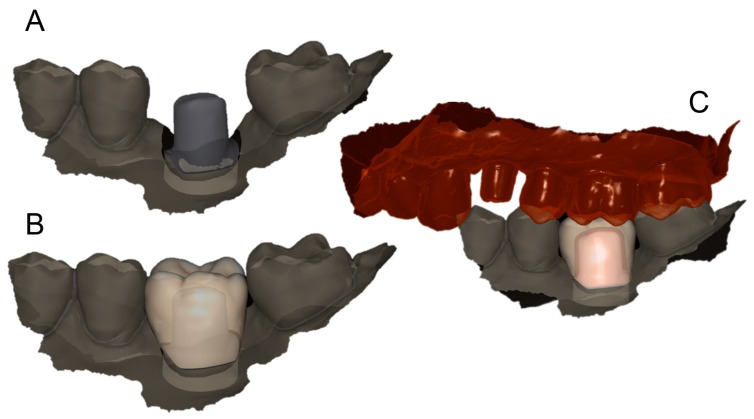
Second CAD scene (PLAN2) with the design of the final restoration. (**A**) the margine line of the individual hybrid abutment is clearly visible, even if subgingival; (**B**) since the margin line is clearly visible and automatically detected by the software, the dental technician can design the final restoration; (**C**) lateral view with detail of occlusion.

**Figure 13 ijerph-16-00317-f013:**
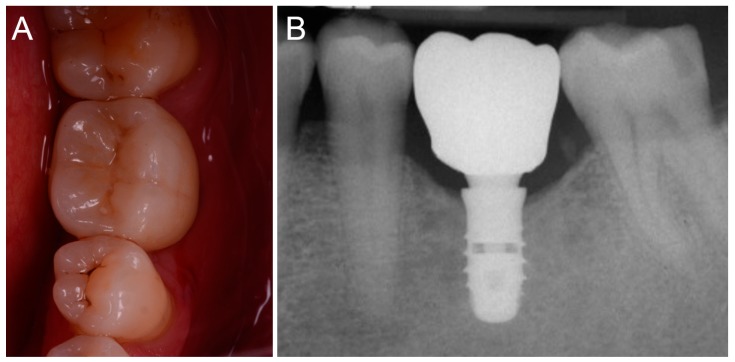
Delivery of the final crown (DONE2). (**A**) clinical view; (**B**) endoral periapical radiograph of the final restoration.

**Table 1 ijerph-16-00317-t001:** Problems encountered at the delivery of the final crowns. Among the problems there were issues with marginal adaptation (MA), occlusal adaptation (OA), interproximal adaptation (IA) and aesthetic integration (IE).

Type of Issue	Incidence	Complication Rate %
*Marginal adaptation (MA)*	0/40 crowns	0%
*Occlusal adaptation (OA)*	2/40 crowns	5%
*Interproximal adaptation (IA)*	1/40 crowns	2.5%
*Aesthetic integration (iE)*	1/40 crowns	2.5%
**Total**	4/40 crowns	10%

**Table 2 ijerph-16-00317-t002:** Failures and complications affecting the implant-supported crowns, occurred during the follow-up period until the 1-year control. Among the complications there were hybrid abutment loss of connection (HALC), zirconia abutment decementation (ZAD), zirconia crown decementation (ZCD).

	N°	Failures	Survival Rate %	Complications	Success Rate %
Gender					
*Male*	12	1/12	91.7%	1/11 ZCD	91.0%
*Female*	13	0/13	100%	1/13 HALC 11/13 ZAD	84.7%
Smoke					
*Yes*	6	1/6	83.4%	0/5	100%
*No*	19	0/19	100%	1/19 ZCD 1/19 HALC 1/19 ZAD	84.3%
Location					
*Maxilla*	25	1/25	96%	1/24 ZCD 1/24 HALC	91.7%
*Mandible*	15	0/15	100%	1/15 ZAD	93.4%
Position					
*Premolar*	12	0/12	100%	1/12 ZAD	91.7%
*Molar*	28	1/28	96.5%	1/27 HALC 1/27 ZCD	92.6%
Titanium base					
*Tibase^®^*	8	0/8	100%	1/8 HALC 1/8 ZAD	75%
*Multitech straight^®^*	10	0/10	100%	0/10	100%
*Multitech angled^®^*	12	1/12	91.7%	1/11 ZCD	91%
Implant diameter					
*3.3 mm*	1	0/1	100%	0/1	100%
*4.1 mm*	15	0/15	100%	1/15 HALC	93.4%
*4.8 mm*	20	0/20	100%	1/20 ZAD 1/20 ZCD	90%
*5.5 mm*	4	1/4	75%	0/3	100%
Implant length					
*6.5 mm*	4	1/4	75%	0/3	100%
*8 mm*	10	0/10	100%	1/10 ZCD	90%
*10 mm*	14	0/!4	100%	1/14 HALC	92.9%
*12 mm*	8	0/8	100%	1/8 ZAD	87.5%
*14 mm*	4	0/4	100%	0/4	100%
**Total**	40	1/40	97.5%	3/39	92.4%

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
