# Peer review of "A Novel Full-Digital Protocol (SCAN-PLAN-MAKE-DONE®) for the Design and Fabrication of Implant-Supported Monolithic Translucent Zirconia Crowns Cemented on Customized Hybrid Abutments: A Retrospective Clinical Study on 25 Patients"

_ijerph, 2019, doi:10.3390/ijerph16030317_

Round 1

Reviewer 1 Report

this is an extremely valuable and interesting and novel work, that presents an entirely digital computer-assisted-design (CAD) technique for the restoration of single implants by means of single crowns supported by customized hybrid abutments. It is of particular interest because it describes a variation of the conventional technique that allow the operator to model the final restoration with all information related to the soft tissue maturation (that were,in the conventional technique, absent) but without losing the quality/details of the margin line. this, simple through a second intraoral scan of the hybrid abutment in position, and without any retraction chord, or the need to displace the soft tissues. i consider this technique of great merit and i strongly suggest the publication of the present work.  

i just am curious about the quality of the superimposition between the original file of the customized hybrid abutment, as modelled in the first CAD, and the captured position of the abutment in the mouth (second scan- the one for the fabrication of the final crown). it is, in fact, key in my opinion, to have the possibility to mathematically understand or calculate the quality of this superimposition, or if there are misfits. can you provide an image that certify this quality? i think to date it is possible, with the use of reverse engineering software (i.e. Geomagic Studio). or as alternative through the same exocad. in any case, the fact that you did not calculate it for each case, the quality of the superimposition, should be considered a limitation of the present study and this should be added in the discussion section of your manuscript.

the fact that the results, in term of clinical fit, were excellent, should be demonstrated also in mathematical terms. i know that it is not easy, but this can be a further advancement in terms of accuracy, for the full digital workflow in dentistry.

in addition, why you did not include in this study fixed partial prostheses (implant bridges of 2-3 elements?). do you believe the present technique is not advisable for longer - span restorations? what are the problems we (may) encounter?

finally, the quality of the english is excellent and does not require editing.

Author Response

this is an extremely valuable and interesting and novel work, that presents an entirely digital computer-assisted-design (CAD) technique for the restoration of single implants by means of single crowns supported by customized hybrid abutments. It is of particular interest because it describes a variation of the conventional technique that allow the operator to model the final restoration with all information related to the soft tissue maturation (that were,in the conventional technique, absent) but without losing the quality/details of the margin line. this, simple through a second intraoral scan of the hybrid abutment in position, and without any retraction chord, or the need to displace the soft tissues. i consider this technique of great merit and i strongly suggest the publication of the present work.  

(authors' reply) thank you very much. yes we feel this paper can introduce a new fully digital technique to improve the quality of the workflow with dental implants. generally, with dental implants, only one single scan is captured with intraoral scanner, i.e. the scan of the scanbody (scanabutment) in position. this allows the clinicians to identify the position of the implant in 3d, and the dental technician to design the provisional as well as the definitive restoration. however, there are two main issues related to this approach. first, the scanbody is not individualized (joda et al. coir introduced the valuable concept of the individualized scanbody technique in digital dentistry, but it is still niche), but generic. it is a cylinder and it does not displace soft tissues like the individual hybrid abutment does. for this reason, positioning the margins of the abutment can be difficult and can give rather unpredictable results, as demostrated by recent papers (pietruscki jk, bmc oral health). in addition, and most important, soft tissue maturation occurs during the provisionalization period and therefore it will be very important to capture this info at the end of the provisionalization period, before to design and fabricate the final restoration. this is key but nobody does it because it would be impossible to capture the subgingival margins of the abutment, and therefore to draw the margin line correctly in addition working on a mesh is worst than working on an original cad design (i.e. nurbs). our approach tries to solve these issues, taking a second oral scan to capture the gingival margins, but superimposing the original cad design file of the hybrid abutment to allow the dental technician to model the final restoration on an original cad (nurbs) and not on a mesh.

i just am curious about the quality of the superimposition between the original file of the customized hybrid abutment, as modelled in the first CAD, and the captured position of the abutment in the mouth (second scan- the one for the fabrication of the final crown). it is, in fact, key in my opinion, to have the possibility to mathematically understand or calculate the quality of this superimposition, or if there are misfits. can you provide an image that certify this quality? i think to date it is possible, with the use of reverse engineering software (i.e. Geomagic Studio). or as alternative through the same exocad. in any case, the fact that you did not calculate it for each case, the quality of the superimposition, should be considered a limitation of the present study and this should be added in the discussion section of your manuscript.

(authors' reply) with the aim of replying to the reviewer's consideration, we have added a new figure (labelled as fig. 11, then we have renumbered all the other figures) in which we use the powerful geomagic studio reverse engineering software, as requested, to exactly calculate the misfit and the error in the exocad superimposition. basically, we have exported the two files (scan of the abutment in position, in mouth and original cad design of the abutment, i.e. as designed in the cad for the provisional) from the exocad, after the superimposition and we calculated the misfit in the reverse engineering. theoretically, a different approach could be to use a reverse engineering all time, to evaluate the mathematical difference; but the use of exocad is more suitable for the dental technicians worldwide. the results were very interesting and we have introduced this in our paper accordingly. one sentence has been introduced in the discussion too, in order to explain this procedure. this procedure has been performed in the selected case only. (please see also supplementary material useful for review).

the fact that the results, in term of clinical fit, were excellent, should be demonstrated also in mathematical terms. i know that it is not easy, but this can be a further advancement in terms of accuracy, for the full digital workflow in dentistry.

(authors' reply) we definitely agree with the reviewer that mathematical data are of paramount importance in dentistry now, and for this reason we have modified our text introducing this control and our figure list and legends accordingly. all changes have been highlighted in the text using a different colour (red)

in addition, why you did not include in this study fixed partial prostheses (implant bridges of 2-3 elements?). do you believe the present technique is not advisable for longer - span restorations? what are the problems we (may) encounter?

(authors' reply) this study only focused on the application with single crowns as it can be considered a pilot study; after the excellent results obtained here, we will start a second study focusing on more complex restorations, such as implant-supported partial fixed prostheses or implant bridges with 3-4 elements. technically, the same procedure can be applied also to longer-span restorations, but this must be clinically validated. 

finally, the quality of the english is excellent and does not require editing.

(authors' reply) thank you very much. our paper has been screened and corrected by a proficient english native speaker from the cambridge proofreading company, therefore the quality of our english is high.

Reviewer 2 Report

this manuscript is innovative but rather technical in nature, and not easy to read for the common dentist, even if well written in english. the topic is of high scientific value because it proposes a novel technique for handling the full digital workflow with single implant crowns. the advantage is to introduce a technique that can reduce errors, mistakes and misfits in the fabrication of single implant crowns via the full digital workflow. 

however some improvements are necessary, in order to help the reader to understand the content. here my few comments to improve the quality of the article:

- abstract, line 26, please refer to the superimposition, you have superimposed the "supplementary abutment design" saved in external folder, exactly over the abutment captured intraorally. please specify that

- i feel the most important outcomes are those measured at the delivery of the final restoration; the 1-year ctrl is of lesser importance here. therefore, i suggest to divide the outcomes into two categories: primary outcomes (those registered at the delivery of the final crowns) and secondary outcomes (the 1-year outcomes). i ask the authors to modify their abstract and text accordingly

- the validity of the present protocol will be fully confirmed, if it is applicable to long span restorations, like partial prostheses. please mention it in the abstract conclusions

- intro, line 136, you should mention that a mesh is derived by a scan - in this case extraoral. it is important to help the reader to understand

- line 140, 146: patented or registered? i think the second, you cannot patent a workflow

- fig. 1, scan 2, there is a typing error please correct

- line 162,164,168 it should read "PLAN2" "MAKE2" and not "CAD2" and "CAM2" please correct

- line 205, please correct the typing error

- line 258, 264, the radiograph is Fig. 12 b and the clinical picture is Fig. 12 a, please correct

- line 319 please set primary and secondary outcome variables

- please use the word "aesthetic" instead of "esthetic" throughout all your manuscript

- line 436, i feel the bonding bases should be sandblasted all times before cementation

- line 685, why you used a different colour?

Author Response

this manuscript is innovative but rather technical in nature, and not easy to read for the common dentist, even if well written in english. the topic is of high scientific value because it proposes a novel technique for handling the full digital workflow with single implant crowns. the advantage is to introduce a technique that can reduce errors, mistakes and misfits in the fabrication of single implant crowns via the full digital workflow. 

(authors' reply) THANK YOU 

however some improvements are necessary, in order to help the reader to understand the content. here my few comments to improve the quality of the article:

- abstract, line 26, please refer to the superimposition, you have superimposed the "supplementary abutment design" saved in external folder, exactly over the abutment captured intraorally. please specify that

(authors' reply) we have modified our abstract as requested.

- i feel the most important outcomes are those measured at the delivery of the final restoration; the 1-year ctrl is of lesser importance here. therefore, i suggest to divide the outcomes into two categories: primary outcomes (those registered at the delivery of the final crowns) and secondary outcomes (the 1-year outcomes). i ask the authors to modify their abstract and text accordingly

(authors' reply) we have modified our text as requested.

- the validity of the present protocol will be fully confirmed, if it is applicable to long span restorations, like partial prostheses. please mention it in the abstract conclusions

(authors' reply) we clearly report the limitations of our study, in accordance with the request of the reviewers.

- intro, line 136, you should mention that a mesh is derived by a scan - in this case extraoral. it is important to help the reader to understand

(authors' reply) we have modified according to the request of the reviewer.

- line 140, 146: patented or registered? i think the second, you cannot patent a workflow

(authors' reply) the reviewer is right, the workflow is registered and not patented.

- fig. 1, scan 2, there is a typing error please correct

(authors' reply) we have modified according to the request of the reviewer.

- line 162,164,168 it should read "PLAN2" "MAKE2" and not "CAD2" and "CAM2" please correct

(authors' reply) we have modified according to the request of the reviewer.

- line 205, please correct the typing error

(authors' reply) we have corrected this error. 

- line 258, 264, the radiograph is Fig. 12 b and the clinical picture is Fig. 12 a, please correct

(authors' reply) we have corrected this error. 

- line 319 please set primary and secondary outcome variables.

(authors' reply) we have set and distinguished between primary and secondary outcomes.. 

- please use the word "aesthetic" instead of "esthetic" throughout all your manuscript

(authors' reply) we have used the term "aesthetics" consistently throughout all text. 

- line 436, i feel the bonding bases should be sandblasted all times before cementation

(authors' reply) we do agree with the reviewer

- line 685, why you used a different colour?

(authors' reply) we have corrected this error. now only the corrections are marked in different colour (red)